# Microstructure Evolution and Mechanical Properties of X6CrNiMoVNb11-2 Stainless Steel after Heat Treatment

**DOI:** 10.3390/ma14185243

**Published:** 2021-09-12

**Authors:** Jia Fu, Chaoqi Xia

**Affiliations:** 1School of Material of Science and Engineering, Xian Shiyou University, Xi’an 710065, China; xia_chaoqi@163.com; 2School of Material of Science and Engineering, Taiyuan University of Science and Technology, Taiyuan 030024, China

**Keywords:** X6CrNiMoVNb11-2 steel, quenching and tempering process, microstructure, chromium carbide precipitation, tempered hardness equation, mechanical properties

## Abstract

X6CrNiMoVNb11-2 supermartensitic stainless steel, a special type of stainless steel, is commonly used in the production of gas turbine discs in liquid rocket engines and compressor disks in aero engines. By optimizing the parameters of the heat-treatment process, its mechanical properties are specially adjusted to meet the performance requirement in that particular practical application during the advanced composite casting-rolling forming process. The relationship between the microstructure and mechanical properties after quenching from 1040 °C and tempering at 300–670 °C was studied, where the yield strength, tensile strength, elongation and impact toughness under different cooling conditions are obtained by means of mechanical property tests. A certain amount of high-density nanophase precipitation is found in the martensite phase transformation through the heat treatment involved in the quenching and tempering processes, where M_23_C_6_ carbides are dispersed in lamellar martensite, with the close-packed Ni_3_Mo and Ni_3_Nb phases of high-density co-lattice nanocrystalline precipitation created during the tempering process. The ideal process parameters are to quench at 1040 °C in an oil-cooling medium and to temper at 650 °C by air-cooling; final hardness is averaged about 313 HV, with an elongation of 17.9%, the cross-area reduction ratio is 52%, and the impact toughness is about 65 J, respectively. Moreover, the tempered hardness equation, considering various tempering temperatures, is precisely fitted. This investigation helps us to better understand the strengthening mechanism and performance controlling scheme of martensite stainless steel during the cast-rolling forming process in future applications.

## 1. Introduction

The hot-rolling composite forming process is a new short-flow near-net-shape-forming process that combines two advanced technologies: the advanced centrifugal casting and fully utilizing the waste heat of casting (without the piercing process, the ring billet consumption can be saved up to 30%) to obtain a high-quality billet. Recently, it has become a new energy-saving technology for manufacturing large-scale aero engine and liquid rocket engine parts, using Fe-Cr-Ni-Mo high-temperature alloy steel and creep-resistant steel. The iron-chromium-carbon ternary alloys of martensitic stainless steel have garnered great interest in recent years due to their superior mechanical properties, such as corrosion resistance, high strength and good toughness, which can be enhanced through high-density nanophase precipitation by heat treatment [1,2]. The Fe-Cr-Ni-Mo alloy is designated as a superalloy martensitic stainless steel (SSC) [3,4,5]. In the heat treatment of the majority of Fe–Cr–Ni ternary and Fe-Cr-Ni-Mo multicomponent alloys, nickel-base intermetallic phases (e.g., NiFe, NiMn, Ni_3_V, Ni_3_Mo, and Ni_3_Nb) are probably formed, and only the metastable phases Ni_3_Mo and the stable Laves phases Fe_2_Mo are formed in Fe-Ni-Mo steels [1,4,5,6,7]. Martensitic stainless steel underwent the evolution of a series of stainless steels, such as: 1Cr13, 2Cr13Mo, 1Cr17Ni2, 00Cr13Ni5Mo, 00Cr14Ni6Mo2AlNb, P91, and 12CrMoV [1,2,3,4,5,6,7]. With its high strength, good hardenability and creep resistance, X6CrNiMoVNb11-2 steel (06Cr11Ni2MoVNb in the CN system) is often used in aero engine turbine discs, large gas turbines, turbine blades, nuclear power equipment, and corrosion-resistant oil pipelines. X6CrNiMoVNb11-2 steel can be strengthened according to the theory that the high-density nanophase precipitation can be promoted by martensite phase transformation through the quenching and tempering process [3,4,8], thus improving the overall performance considering the coarse grain sizes [9,10,11] and heat-treatment conditions [12] when casting.

In terms of martensitic stainless steel, a solid-solution thermal treatment process has a great influence on microstructure and the mechanical properties of high-strength steel, and the grain size after the two-time tempering process is finer [12]. In terms of corrosion resistance, increasing the molybdenum content can significantly reduce the corrosion reaction rate [13]. The impact toughness of 00Cr13Ni4Mo steel, after solid solution at 1040 °C for 1 h and two-time tempering at 600 °C for 3 h, was greatly improved [14]. In fact, the applications for aerospace and other corrosive environments seem to largely overlap with welding parts from stainless steel, where diffusion bonding is used [15]. In terms of 13Cr steel, lath martensite and a small amount of retained austenite were obtained after quenching at 800–1100 °C for 30 min [16]. However, with an increase in austenitizing temperature, the martensite lath and retained austenite grains became coarser [17,18]. Therefore, after an austenitic stabilization treatment, the quenching and tempering processes of low-carbon martensitic stainless steel become meaningful in practical applications [6,7,8], especially in terms of X6CrNiMoVNb11-2 steel with excellent hardenability.

In previous research, a precise constitutive model, considering strain compensation and the hot processing map of 06Cr11Ni2MoVNb martensitic stainless steel, has been established to distinguish the “safety zone” from the “unstable zone” [19,20]. The hot deformation parameters of ring-rolling processing have been optimized to investigate the deformation rules of disk pieces [21]. However, for this kind of Fe-Cr-Ni-Mo martensitic stainless steel, research on the heat-treatment process of X6CrNiMoVNb11-2 steel only focuses on conventional annealing, quenching, tempering and nitriding treatments [21,22,23,24,25,26], the influencing rule of the cooling medium on yield strength, tensile strength, elongation and impact toughness during quenching and tempering has not yet been identified. Hence, the current research on mechanical properties under different cooling conditions by means of a mechanical property test is very limited [27,28,29]. Based on the initial X6CrNiMoVNb11-2 steel of the casting-rolling disk, achieved by pre-heat treatment and the hot-rolling process, the identification of microstructure evolution and mechanical properties is urgently needed to reveal the reinforcement mechanism of phase transformation after the heat-treatment process (quenching at 1040 °C and tempering at 300–670 °C). Therefore, the corresponding mechanical properties are tested with comparisons under various heat treatments, and the results will have important theoretical value and practical implications for performance control.

## 2. Materials and Methods

### 2.1. Sample Preparation and Initial Properties

#### 2.1.1. Casting Process and Chemical Composition of X6CrNiMoVNb11-2 Steel

The casting process was conducted with an electric arc furnace, using the method of non-vacuum induction + electro-slag remelting and smelting, carried out by smelting laterite nickel ore with 1.5–3% nickel content into nickel-bearing pig iron in a blast furnace. In step (1), the laterite nickel ore was crushed, sieved and dried to obtain ore fines and blocks within a certain size range, during which process, coke was used as the fuel and limestone was added as a flux. Then, the mixed ore was put into the sintering machine to form a sintering ore, and then put into the blast furnace to produce nickel-bearing pig iron. In step (2), using an electric arc furnace, an induction furnace and a hot blast furnace, the nickel-containing pig iron was melted into nickel-containing molten iron or semi-molten steel. Step (3) involves decarbonization through the argon–oxygen decarburization (AOD) process to achieve molten iron or semi-molten steel. In step (4), after AOD slagging, the molten steel was put into an LF furnace with synthetic slag added, and then the alloy composition was adjusted. In the final step (5), heating and argon-blowing refining were carried out to deoxidize, desulfurize and remove other harmful impurities. The Nb-based secondary carbides/nitrides phases need very high austenitizing temperatures, due to the solidification solution of the Nb element with a high melting point. The ingot was made of pure Fe and high-purity Cr, Ni, Co, Mo, and Mn elements, with a weight of 200 kg.

#### 2.1.2. The Hot-Rolling Process and the Initial As-Rolled Mechanical Properties

Before the hot-rolling process, the pre-heat treatment consisted of two steps: (1) a homogenizing treatment, consisting of heating the steel to 700 °C, holding it for 5–6 h in the box resistance furnace, and then cooling it to room temperature; (2) a normalizing treatment, consisting of heating to 1040 °C, holding it for 1–2 h, and then cooling to room temperature. For conducting a scaling test of the hot-rolling process, the as-cast ring ingot (50-mm inner diameter, 280-mm outer diameter, and 32 mm in height) was heated to 1250 °C and homogenized for 4 h, then cooled to 1200 °C for 1 h, this temperature being equivalent to the as-cast state for hot rolling. The designed cone roll die was loaded on the ring rolling tester to finish the isothermal hot-rolling procedure. The final size of the as-rolled disk is 135-mm inner diameter, 386-mm outer diameter and 28 mm in height. Through pre-heat treatment and the hot-rolling process, the as-rolled disk was obtained based on the as-cast ring ingot. The mechanical properties of the as-cast ring ingot and initial as-rolled disk were measured and are listed in Table 1.

From Table 1, the tensile strength, yield strength, elongation and impact toughness of as-rolled specimens were increased by 110 MPa, 40 MPa, 3.8% and 10 J/cm^2^, compared with as-cast properties that were provided by the vendor of the Northeast Special Steel Co. Ltd. The initial as-rolled disk was used for preparing a heat treatment experiment, as given in Section 2.2.

### 2.2. Heat Treatment Schedule and Properties Measurements

#### 2.2.1. Experimental Schedule of the Heat-Treatment Process

The as-rolled disk was used to prepare specimens with a DK7740 wire cutting machine (Jiangsu Fangzheng CNC Machine Tool Co., Ltd., Taizhou, China). Cylindrical samples of 50 ± 0.05 mm in diameter and 90 ± 0.05 mm in length were used. The schedule of the heat treatment experiment is shown in Table 2.

From Table 2, the quenching process involves heating samples to 1040 °C with a holding time of 1 h, through oil cooling, air cooling, furnace cooling, or the PAG medium cooling. Meanwhile, the tempering process was carried out at 300–670 °C with a holding time of 1.5–2 h, and then the samples were cooled by air or the oil medium. Samples were heated by a KF1600 box resistance furnace, with a heating rate of 7 °C/min. Four kinds of quenching mediums (oil, air, furnace, and polyalkylene glycol (PAG) quenching liquid) and two tempering mediums were used to study the effect of each cooling mode and holding time on the mechanical properties of heat-treated samples.

#### 2.2.2. Experimental Measurement of Mechanical Properties

The DIL805A/D differential dilatometer (Waters corporation, Delaware, USA) was used to measure the linear expansion curve of the sample (Φ 4 mm × 10 mm), to record the relative displacement and to determine the temperature at the characteristic points of phase transition, such as M_s_, M_f_, A_c1_ and A_c3_. Besides this, the V-notched Charpy notched specimen for toughness measurement was tested, with the size of 5 mm × 10 mm × 55 mm (GB/T229-1994). Tensile strength was tested, with the flat section size seen in ASTM-E8M (the specifications being a gauge length of 50.0 ± 0.1 mm, contraction width 12.5 ± 0.2 mm, thickness of the tensile specimen 4.0 mm, bending radius ≥ 12.5 mm, contraction length ≥ 60 mm, clamping length ≥ 75 mm, clamping width 20 mm). The hardness was measured by the Vickers hardness tester (the load is 200 gf, the dwell time, about 20 s, the spacing between indentations, about 0.1 mm). Each specimen was tested three times and then averaged. After heat treatment, each specimen was polished and etched with the etchant (4 mL HF + 4 mL HNO_3_ + 92 mL distilled water) for about 5 min. Phases and microstructures were observed and analyzed by means of an XRD-6000 X-ray diffractometer (Shimadzu Corporation, Tokyo, Japan), VHX-600E optical microscope (OM) (KEYENCE Co. Ltd., Osaka, Japan), JSM-7100F scanning electron microscope (SEM) (JEOL Ltd., Tokyo, Japan) with Oxford AZtecX-Max20 energy spectrum probe (Oxford Instruments, Oxford, UK) for the energy dispersive spectroscopy (EDS) analysis, JEM-2100Plus transmission electron microscope (TEM) (JEOL Ltd., Tokyo, Japan). The hardness, elongation and impact toughness were carried out on the FM-700/SVDM4R microhardness tester (Future Tech Enterprise Inc., New York, USA), the FLFS-105 slow strain-rate tensile tester (FULE Instrument Technology Co. Ltd., Shanghai, China) and the C64-305 MTS universal tester (MTS Systems (China) Co. Ltd., Minnesota, USA), respectively.

## 3. Results and Discussion

### 3.1. Analysis of the Alloy Phase Diagram and TTT/CCT Curve

Chemical composition is: C: 0.075%, Cr: 11.390%, Ni: 1.570%, Mo: 0.390%, Nb: 0.08%, V: 0.150%, Mn: 0.380%, Si: 0.180%, Ti: 0.060%, N: 0.020%, P: 0.015%, S: 0.004%, other: Fe. Thus, the chromium and nickel equivalents by Schaeffler diagram [11] are calculated:Cr_Equ_ = %Cr + %Mo + 1.5 × %Si + 0.5 × %Nb = 11.39% + 0.39% + 1.5 × 0.18% + 0.5 × 0.08% = 12.09%(1)
Ni_Equ_ = %Ni + 30 × %C + 0.5 × %Mn = 1.57% + 30 × 0.075% + 0.5 × 0.38% = 4.01%(2)
where Cr_Equ_ and Ni_Equ_ are the chromium equivalent and the nickel equivalent, respectively. Figure 1 shows the Schaeffler diagram and equilibrium phase diagram.

Figure 1a is used to predict stable phases at room temperature during the solidification process [30]. As the values of Cr_Equ_ and Ni_Equ_, it has guaranteed the obtaining of a martensitic from the hardening treatment. Form Figure 1b, within the mass percent of carbon less than 0.1%, it can be inferred in Figure 1b that martensitic phase mainly exists [11,22], equilibrium phase is about α + γ + (Nb,Cr)(C, N) + Cr_2_N + M_23_C_6_ [30].

#### 3.1.1. Analysis of the Alloy Phase Diagram

The solidification alloy phase diagram by JMatPro software is shown in Figure 2.

The alloy phase diagram in Figure 2a is composed of the liquid zone, the ferrite zone and the austenite zone, containing ten phase zones of alloy compounds, such as MS_B81, M(C, N), MnS, M_23_C_6_, M_3_P, Laves, Alpha-Cr, G-Phase, M_2_(C, N) and M_6_C. When the temperature reaches 778.80 °C, austenite begins to appear; thus, A_c1_ = 778.80 °C, A_c3_ = 791.20 °C. The pouring temperature should be above the liquid point at 1449.5 °C, and the austenite zone is between 815 °C and 1201 °C. From Figure 2b, with the increase of the temperature from 300 to 1100°C, the thermo-physical parameters including the thermal expansion coefficient, thermal conductivity and specific heat all show an increased tendency. The thermal expansion coefficient, thermal conductivity and specific heat at room temperature are calculated to be 21.28 × 10^−5^/°C, 59.24 W/m °C and 59.53 J/Kg °C.

#### 3.1.2. Analysis of the Phase Composition and CCT Curve

The phase composition and CCT curve are calculated to show heat treatment parameters, considering the martensite phase transition, in Figure 3.

From Figure 3a, after quenching, the austenite phase changes to the martensite phase as the transformation of the austenite phase begins at 778.80 °C. It should also be noted that, in nano-precipitation, it is supposed that all primary secondary coarsened phases (carbides, nitrides, and carbonitrides) obtained from the solidification are completely solved in the austenitizing stage, leading to the obtaining of homogenous austenite [31]. From Figure 3b, the martensite transformation starts (M_s_) when the temperature drops below 290.30 °C and finishes when the temperature reaches about 140.21 °C, under a cooling rate of 1.0 °C/s. During cooling from 253.80 °C to 169.00 °C, the martensite phase fraction increased from 50% to 90%.

### 3.2. Phase Transformation Temperatures and Martensitic Kinetic Equation

#### 3.2.1. Determination of Martensitic Phase Transition Point

The martensite transformation start temperature (M_s_) [32], the austenite start temperature (A_c1_) [33] and the austenite finish temperature (A_c3_) [34] are calculated as:M_s_(°C) = 551 − 462.0(%C + %N) − 9.2(% Si) − 8.1(% Mn) − 29.0(%Ni) − 13.7(%Cr) − 18.5(% Mo)(3)
Ac_1_(°C) = 723 − 10.7(%Mn) − 16.9(%Ni) + 29.1(%Si) + 16.9(%Cr)(4)
Ac_3_(°C) = 871 − 254(%C^1/2^) − 14.2(%Ni) +51.7(%Si).(5)

Thus, M_s_, A_c1_ and A_c3_ are 293.59 °C, 724.67 °C and 863.90 °C, respectively. The austenitic transformation continues up to 790.31 °C. Combined with the linear expansion curve, the phase transition temperatures are finally determined by means of the single tangent calculation [34]: M_s_ = 292.71 °C, M_f_ = 133.95 °C. The difference of M_s_ between the reference value [32] and the single tangent calculation value is due to the fact that the element Nb generally decreases the M_s_ temperature [10,31,32].

For the determination of the M_s_ temperature, when the sample is heated to 1040 °C (10 °C/s) and remains at this temperature for 30 min, the length increases due to the thermal expansion of the austenite phase and the dissolution of the precipitate, if any [20]. Figure 4 shows that the linear expansion curve, with a cooling rate of 1.0 °C/s, depends on the temperature of the specimen (Φ 4 mm × 10 mm), compared with the simulated curves.

The A_c1_, A_c3_, M_s_ and M_f_ can be determined through the verification of the experimental phase transition point [31,32]. Figure 4a shows the transition temperatures (A_c1_ = 774.77 °C, A_c3_ = 851.21 °C, M_s_ = 292.71 °C and M_f_ = 133.95 °C) from the measured length of the sample by the dilatometer, by heating to 1040 °C for 30 min and cooling at a rate of 1.0 °C/s, which are close to the values of A_c1_ = 774.00 °C, A_c3_ = 861.00 °C, as taken from reference [35]. Since the specific volume of austenite is smaller than that of martensite, the rapid expansion of the specimen at the M_s_ point during the transformation from austenite to martensite is observed near 292.71 °C. According to Figure 4a, the sample length firstly increases linearly and then decreases, indicating an austenite transformation at the inflection. The phase transformation from austenite to martensite happens during the quenching process at a cooling rate of 1.0 °C/s. Figure 4b shows the comparison of the experimental and simulated curves; the practical M_s_ is lower than that of the simulated one. The simulated M_s_ point trends to be consistent, from various initial austenitizing temperatures of 1040–1100 °C.

#### 3.2.2. Calculation of the Phase Transformation Kinetic Equation

The kinetics of austenite and martensite transformation can be expressed by the Johnson–Mehl–Avrami (JMA) equation and the Koistinen–Marburger (KM) equation [36]:(6)$$\xi_{A}=1-\exp(-kt^{n});\;\xi_{M}=1-\exp(-\alpha (M_{s}-T_{q}))$$
where the amount of new phase formation is expressed as ξ, k is the rate constant (related to the phase transition temperature, free energy, and other parameters), t is the time, *n* is the time index (related to the nucleation and growth mechanism), and the current temperature is T_q_. Using the thermal expansion curve, the temperatures of the beginning time and the ending time of the phase transition can be directly marked and determined by using the lever theorem, and then the phase variable in Equation (6) can be calculated.

Phase transformation curves at a cooling rate of 1.0 °C/s are obtained and are shown in Figure 5.

As seen in Figure 5, the kinetic parameters of the phase transition under specific reaction conditions by the linear fitting of phase transition curves are expressed. For the austenite volume fraction, ln[−ln(1 − ξ)] = ln(−k) + *n*·lnt. The martensitic volume fraction decreases non-linearly with the increase in temperature. The data of ln(1 − f) and M_s_ − T_q_ are linearly fitted, to obtain *n* and α for the JMA and KM equations at a cooling rate of 1.0 °C/s.

The austenite and martensite transformation kinetic equations are obtained as:(7)$$\xi_{A}=1-\exp(-0.08t^{1.95});\;\xi_{M}=1-\exp(-0.0198(M_{s}-T_{q}))$$

### 3.3. Analysis of SEM, XRD, EDS and Phase before and after Heat Treatment

Microstructures under various heat treatment conditions are shown in Figure 6.

From Figure 6a,c, the mixture consists of the approximately parallel lath martensite and the delta ferrite, with a morphology of isolated islands that are mainly located at grain boundaries. Besides this, the SEM shows that the size of the Nb particle is undistributed and uneven, within a length size of 742.9 nm–6.557 μm. In this steel, niobium is expected to combine with most of the carbon to form niobium-containing carbide, which effectively hinders the retained austenite from being present in the microstructure after the quenching treatment. From Figure 6b,d, the microstructure, after quenching and tempering, is formed of tempered martensite and carbide. Thus, austenitizing at 1040 °C for 30 min and then quenching in oil can ensure that the austenite is mainly transformed into lath martensite, but this is not sufficient to completely dissolve the carbides [28,37,38]. Moreover, the distribution of Nb(C,N) in Figure 6a is not uniform, in comparison with the microstructure seen after the quenching and tempering processes in Figure 6b. In Figure 6d, chromium carbide M_23_C_6_ precipitation can be observed, with cubic NiFe high-density co-lattice nanocrystalline precipitation during the tempering process at 650 °C.

#### 3.3.1. SEM and XRD Analysis before and after Heat Treatment

Taking the quenching at 1040 °C for 1.0 h + tempering at 650 °C for 1.5 h by air cooling as an example, the microstructures formed under various quenching processes can be seen in Figure 7.

The quenching microstructure from Figure 7a is mainly lath martensite, while the tempering microstructure in Figure 7b,c is mainly tempered martensite + carbide. The microstructure under processing (quenching oil cooled at 1040 °C and tempered at 650 °C for 1.5 h) shown in Figure 7c is more uniform. Compared with the quenching air-cooled microstructure in Figure 7b, the quenching oil-cooled microstructure in Figure 7b is smaller and more uniform, due to the different cooling rates. The X-ray anode source for XRD measurements is Cu k-alpha, and the XRD curves under various processes are shown in Figure 7d. The peak positions are marked at 110 bcc, 200 bcc and 211 bcc at nearly 44.50°, 65.00° and 82.50°, respectively, showing the main martensite matrix microstructure. There are also four types of MX particles (such as VX, NiX, NbX and CX [39] in Figure 7d) observed in the XRD of quenched and tempered samples, compared with fewer types of MX particles in the quenched sample in Figure 7d. After heat treatment, the average austenite diameter is 25 µm, corresponding to the ASTM-8 grain size.

#### 3.3.2. SEM-EDS Analysis before and after Heat Treatment

SEM-EDS analysis of X6CrNiMoVNb11-2 steel before and after heat treatment is shown in Figure 8, with the same heat treatment condition as that in Figure 6.

From Figure 8a,b, the carbide formed by ferrochromium is distributed along the grain boundaries. This is due to the fact that the difference in the saturated solubility of the trace carbon in the γ-phase and α-phase results in the continuous precipitation (above A_c3_) of carbon from the phase during a slow cooling process. In fact, alloy elements of Mo, V and Nb can also effectively prevent dislocation from over-aging; thus, they can hinder the decomposition of martensite and enhance the tempering resistance [14]. In addition, the precipitation of different nano-alloy phases can also have an over-aging effect and inhibit the decomposition of martensite, thus improving the anti-tempering performance. It can be seen from Figure 8c that the coarse residual NbX particles are also present. Thus, high-melting-point Nb should be strictly controlled since the as-cast ingot contains niobium carbonitride in the manufacturing procedure specification, used in the production of special N-alloyed steels. Besides this, the martensite morphology is similar to that of the 12% Cr steel observed by Pryds and Huang [39]. As is shown in Figure 8d, during the tempering process, tiny pieces of VX and spherical NbX are formed. The NbX morphology is characterized by a grain boundary within the shape of the allotropy and the secondary sawtooth, where martensite is reticularly distributed along the ferrite grain boundary [40]. As the Nb element has a stronger affinity to C and N than Cr, the addition of Nb will lead to the precipitation of Nb carbides (and nitrides) [39], which will reduce the formation of Cr carbides and CrN nitrides [30,40,41].

### 3.4. Phase Analysis before and after Heat Treatment

#### 3.4.1. EDS Analysis before and after Heat Treatment

The results of EDS analyses before and after heat treatment are shown in Figure 9.

As seen in Figure 9a, the distribution of Nb(C, N) is approximately equal to that of Nb(C, N) in the carbonitride of martensite in quenched and tempered steel, as seen in Figure 9b. During the tempering process, cubic chromium carbide M_23_C_6_ is precipitated from the martensite matrix, which reduces the concentration of chromium in martensite [42,43]. Meanwhile, Mo, V and Nb elements can inhibit the recovery of the dislocation substructure in martensite and promote the increase in the nucleation location of the precipitated strengthening phase. These elements make the refined grains more uniform and thus can greatly increase strength, improve toughness and reduce damage.

The EDS spot analyses before and after heat treatment are listed in Table 3.

As can be seen in the analysis of metallographic and EDS, the microstructure is mainly martensitic after the quenching process, with a certain content of Nb(C, N) phases. According to the EDS analysis, the elements Nb, C, Fe, and Cr are present at the spot positions in smaller particles, as seen in Table 3. It can be concluded from references [44,45,46] that Nb(C, N), Fe_2_Mo (Laves), Fe_2_Nb and Fe_3_Nb_3_C (M_6_C) particles can also be formed in Nb-containing ferritic stainless steel. These particles are assumed to be NbC, and the smaller particles are usually arranged in a linear fashion around dislocations or grain boundaries [47]. Most elements (Mn, Ti, Cr, Nb, etc.) can significantly reduce the M_s_ point value [46,47,48].

#### 3.4.2. Phase Analysis before and after Heat Treatment

The phase analysis of the microstructure after various heat treatments is shown in Figure 10.

As is seen in Figure 10a, there are long granular precipitates with a length of about 80 nm and a width of about 20.0 nm, which are smaller than the spherical Laves-Fe_2_Mo phase. The carbon precipitates from the phase form carbides with the surrounding Cr and Fe, which are distributed along the grain boundary [30,43,46]. From Figure 10b, the size of the acicular martensite is at submicron level, as the martensitic needles formed are tens of microns in size, which shows similar acicular martensite to that in the report by Krakhmalev [21], but these needles are much smaller in size than that reported by other researchers [49,50]. As is seen in Figure 10b, a high-density co-lattice nanocrystalline precipitation phase (NiFe, Ni_3_V, Ni_3_Nb, etc.) is observed, which can greatly increase the impact toughness during the tempering precipitation process [1,2]. As is shown in Figure 10c, more martensite phases are formed after oil quenching, and the morphology of the grains is completely changed, from 27.4 nm to 62.8 nm, after the tempering process. Figure 10d shows the bright-dark field image and selected diffraction pattern of a quenched-tempered specimen. The fine nano-sized Ni_3_Mo and Ni_3_Nb precipitates are mainly dispersed and are rod-shaped and oriented, with a size of about 30 nm. For the martensite matrix α and the reversed austenite γ’, there is an apparent parallel orientation in relation to ${\left(\bar{1}11\right)}_{\gamma^{\prime }}∥{\left(\bar{1}10\right)}_{\alpha }$, ${\left(\bar{1}12\right)}_{\gamma^{\prime }}∥{\left(\bar{1}\bar{1}3\right)}_{\alpha }$, ${\left(0\bar{2}2\right)}_{\gamma^{\prime }}∥{\left(002\right)}_{\alpha }$, which is consistent with the K-S relation. The content of the reversed austenite γ’ is very low. It is should be noted that Ni_3_Mo is rod-shaped, where the length direction of the rod and matrix is in the <111> direction [2]. The close-packed surface and direction of δ-Ni_3_Mo and η-Ni_3_Nb phases are parallel to the matrix, with the existence of the coherent or semicoherent relation between δ-Ni_3_Mo/η-Ni_3_Nb and matrix, thus the strength is remarkably improved [1,2,51,52,53,54].

The phases of X6CrNiMoVNb11-2 steels after various heat treatments are shown in Table 4.

As can be seen in Table 4, the softening behavior occurs after quenching and tempering, which may be due to the precipitation of the coarse-grained M_7_C_3_ carbide at 550 °C and its partial transformation from M_7_C_3_ carbide into the new M_23_C_6_ carbide precipitates at 650 °C. From their size and distribution, it can be inferred that smaller particles are probably formed in the solid state by precipitation [1,41]. It is worthy of notice, in Figure 6, Figure 7, Figure 8, Figure 9 and Figure 10, that the crystalline sizes under condition I (initial state), II (quenching at 1040 °C in oil) and III (quenching at 1040 °C in oil + tempering at 650 °C), as determined by SEM and EDS, are 66.2 nm, 27.4 nm and 62.8 nm, respectively. The EDS results show that the carbides at 550 °C and 650 °C are M_7_C_3_ and M_23_C_6_, which is in accordance with the findings that carbides at 200 °C, 550 °C and 650 °C are found to be M_3_C, M_7_C_3,_ and chromium-rich M_23_C_6_, similar to martensite stainless steel as reported by Calliariet [48].

#### 3.4.3. Hardness Analysis after Various Heat Treatment Processes

The tempering hardness H (HRC), considering two factors of tempering temperature T (°C) and tempering holding time t (h), can be expressed as [55]:(8)$$H_{Hardness}(HRC)=H_{Time}^{}+H_{Temperature}^{}=f_{1}(t,H_{1})+f_{2}(T,H_{2})$$

In the equation, *H_Hardness_* is the overall hardness equation of the quenched steel, *H_Time_*(*HRC*)* = f_*1*_*(t,* H_*1*_*)** is the partial hardness equation caused by changes to the tempering time, and *H_Temperature_*(*HRC*)* = f_*1*_(*T,* H_*2*_*)** is the partial hardness equation caused by the tempering temperature. The effect of two quenching mediums of oil cooling (O.C.), air cooling (A.C.), and different tempering temperatures (630–670 °C) on the Rockwell overall hardness value are mainly considered.

Meanwhile, the tempering hardness curves under different cooling mediums by the measured hardness data are obtained and shown in Figure 11.

From the dotted line of Figure 11a, each colorful scatter point represents the averaged hardness value when testing each specimen 5 times. It can be seen from Figure 11b that with the increase in tempering temperature within the range of 300 to 670 °C, the hardness decreases greatly. When the tempering temperature is above 600 °C, most of the reverse austenite will change into quenched martensite during the cooling process, due to the decrease in the stability of the reverse transformed austenite [56]. Besides this, from the microstructure analysis, the grain size after tempering at 650 °C for 1.5 h is more uniform and finer than that after tempering at the same 650 °C for 2 h. The quenched martensite shows high-density dislocation, which gives the martensitic stainless steel the characteristics of high strength and low plasticity [12,57]. Therefore, when above 600 °C, the hardness gradually increases, due to the formation of quenched martensite. From Figure 11b, under the same quenching treatment, the highest hardness appears when tempering at 350 °C, and the lowest value is at 670 °C.

Considering the different heat treatment schemes (Q_Air + 1.5 h T.Air, Q_Air + 2.0 h T.Air, Q_Oil + 1.5 h T.Air and Q_Oil + 2.0 h T.Air), the hardness-temperature (H-T) fitting curves under different tempering processes at various temperatures are obtained as follows:(9)$$H(HRC)=\left\{ \begin{array}{l} 24.99+0.11T-(1.45\times 10^{-4})T^{2}\;(R=0.91,\;Q.\mathrm{Air}1.5\;h\;T.\mathrm{Air})\\ 31.71+0.09T-(1.40\times 10^{-4})T^{2}\;(R=0.92,\;Q.\mathrm{Air}+2.0\;h\;T.\mathrm{Air})\\ 24.71+0.10T-(1.34\times 10^{-4})T^{2}\;(R=0.94,\;Q.\mathrm{Oil}+1.5\;h\;T.\mathrm{Air})\\ 20.11+0.13T-(1.73\times 10^{-4})T^{2}\;(R=0.93,\;Q.\mathrm{Oil}+2.0\;h\;T.\mathrm{Air}) \end{array}\right.$$
where T is the tempering temperature (300 °C ≤ T ≤ 670 °C), Q.Air represents the quenching process with air-cooling, Q.Oil stands for the quenching process with oil-cooling, T.Air is the tempering process with air-cooling and R is the correlation coefficient of the fitting curves after various quenching and tempering processes. The four R values show the high correlation between the equation and the experimental data, with 0.91, 0.92, 0.94 and 0.93 by the Q.Air + 1.5 h T.Air, the Q.Air + 2.0 h T.Air, the Q.Oil + 1.5 h T.Air and the Q.Oil+ 2.0 h T.Air processes, respectively.

In this study, the influences of the tempering temperature and the tempering time are taken into account, and the final fitting equation of the tempering hardness curve is:(10)$$H(HRC)=25.38+0.11T-(1.48\times 10^{-4}{)T}^{2}\;(300\;{}^{\circ}C\leq T\leq 670\;{}^{\circ}C).$$

Similarly, R is calculated to be 0.91, showing a high correlation between the measured tempering hardness data and the calculated hardness data. Based on the absolute difference between the calculated tempering hardness value and the measured value, the standard deviation S is calculated to be 1.49; thus, the reliability of the tempering hardness equation is verified.

### 3.5. Experimental Mechanical Properties under Various Heat Treatments

#### 3.5.1. Effect of the Quenching Cooling Rate on Properties

When the quenching temperature of martensitic stainless steel is low, not only the martensite exists in the quenched microstructure but also a proportion of untransformed ferrite in the quenched martensite matrix, which reduces the mechanical properties of the quenched specimen. Meanwhile, when the quenching temperature is too high, the austenite grains will be coarsened, resulting in coarser martensite [51]. Therefore, a quenching temperature should be selected within an appropriate range, neither too low nor too high, or the expected requirements will not be met [41,52]. Then, the effect of quenching cooling rate on the mechanical properties during quenching at 1040 °C and tempering at 650 °C is investigated through the analysis of the measured data.

The ultimate tensile stress (maximum stress on the stress–strain curve), yield stress (stress at 0.2% offset strain) and impact toughness are measured and listed in Table 5.

In Table 5, the Oil., Air., Fur., and Con. stand for the oil cooling, air cooling, furnace cooling and constant cooling speed of 1.0 °C/s, respectively. The quenching cooling rate has a greater impact on the impact toughness, and the quenching process by oil cooling has the highest impact toughness. The faster the cooling rate is, the higher the impact toughness is. With the decrease in the quenching cooling rate, the tensile strength and the yield strength also decrease, due to the existence of the retained austenite in both air-cooled and oil-quenched specimens [16,18,54].

Results also show that strength decreases with the decrease in the quenching cooling rate because of carbide precipitation and temper embrittlement. Commonly, for different parts (the outer side and the center position) of the test steel, there is not much difference in the strength and the toughness when under the same treatment process. This is due to the fact that most of the specimen sizes are within the critical quenching radius.

#### 3.5.2. Effect of Tempering Cooling Rate on Properties

Samples are quenched at 1040 °C in oil or other mediums first, then tempered at 650 °C. The effect of tempering cooling rate on mechanical properties is shown in Table 6.

It can be seen from Table 6 that different tempering cooling rates have little effect on strength. Conversely, the quenching cooling rate has a great effect on strength, which is similar to that on impact toughness. Thus, with a decrease in the quenching cooling rate, the impact toughness shows a downward tendency. As can be seen in Table 6, the impact toughness decreases with the decrease in the quenching and tempering cooling rate; thus, high impact toughness can be obtained by quenching and tempering with a higher cooling rate at the same time.

It can be seen from the different quenching processes shown in Table 5 and various quenching and tempering processes shown in Table 6 that the quenched and tempered specimen using the oil medium with a high cooling rate show a higher impact on toughness than other mediums.

#### 3.5.3. Comparison of Hardness under Various Processes

The hardness of the quenched and tempered specimen is within the required range of 300–340 HV. For different cooling mediums, the surface hardness and the core hardness by quenching at 1040 °C and tempering at 650 °C are shown in Table 7.

As can be seen in Table 7, it can be concluded that the cooling ability of the steel core is insufficient when quenching with oil, which results in lower hardness at the central core without meeting the engineering requirements for the intended application, and the hardness difference between the surface and center is about 27 HV. The tempered hardness of the steel core can reach the standard requirement when the PAG polymer medium is used for quenching and tempering; however, the hardness difference between the surface and center is still large, due to the precipitation of carbide during the tempering process [53,54].

### 3.6. Comparison of Mechanical Properties under Various Processes

#### 3.6.1. Determination of the Tempering Hardness Equation

The measured mechanical properties, as well as grain size under various quenching temperatures of 1020–1060 °C by oil cooling and the subsequent tempering temperatures of 630–670 °C by air cooling, are shown in Figure 12.

From Figure 12a, with an increase in quenching temperature from 1020 to 1060 °C, the yield strength and tensile strength increase. However, with an increase in tempering temperature from 630 to 670 °C, the diffusion of carbon in martensite leads to the decrease in carbon content in martensite; as a result, the tensile strength and yield strength show a decreasing trend. From Figure 12b, the quenching temperature has little effect on plasticity, but it does have a great effect on the grain size and impact toughness. As the austenite grain size is fixed during austenitizing, the quenching temperature (1020–1060 °C) has a great influence on the initial austenite grain size, while the tempering temperature (630–670 °C) has little influence on it. At the same quenching temperature, the tempering temperature has little effect on the impact energy, with only slight changes. The tempering temperature has little effect on the plasticity of the reduction of the area and the elongation. At different quenching temperatures, the plasticity reaches its optimum value when tempered at 650 °C. In terms of the comparison of strength, plasticity and impact properties, X6CrNiMoVNb11-2 steel has the best comprehensive properties when quenched at 1040 °C and tempered at 650 °C. Therefore, according to the measured mechanical properties, the ideal tempering temperature is determined to be 650 °C, with a hardness of about 308–340 HV, as can be seen in Table 7.

#### 3.6.2. Comparison of Tensile Properties under Various Processes

Mechanical properties, hardness and the subsequent phases under different heat treatment processes (quenching at 1040 °C and tempering within the range of 550 to 670 °C for 1.5 h with air cooling) with two quenching conditions are listed in Table 8.

As is seen in Table 8, the process of oil quenching at 1040 °C and tempering at 550–670 °C shows good ductility, with elongation over 15.5% and the reduction of area over 48%. Compared with the as-rolled properties in Table 1, except for a slight decrease in elongation, yield stress, the ultimate tensile stress, hardness and impact energy are sharply increased after the quenching process. When the tempering temperature rises from 550 °C to 650 °C, the tensile strength decreases and the elongation increases gradually. Compared with the tempering temperature at 550 °C, the tensile strength, yield strength and impact toughness decrease slightly, but the elongation increases significantly up to 17.9% after quenching at 1040 °C and tempering at 650 °C with air cooling. The quenched and tempered specimen using the oil-cooling medium, with a high cooling rate, has a higher yield strength, higher impact toughness and lower tensile strength values than that using the air-cooling medium. Conversely, with the increase in tempering temperature, the yield stress, the tensile stress and hardness decrease slightly, while the elongation and the impact energy increase. The elongation and impact toughness are the best when using the process of quenching at 1040 °C by oil cooling + tempering at 650 °C for 1.5 h with air cooling, showing the best comprehensive performance.

In all, the composite-forming process as a new technology where the large ring parts are formed by hot rolling, based on the as-cast billet, keeps gaining attention. The shape-controlling technology of the casting-rolling forming process tends to be perfect [30,58], while the performance-controlling technology is still in a technical bottleneck [25,26]. From a research perspective, except for a heat treatment investigation in the casting-rolling forming process, the grain refinement mechanism during plastic deformation and the performance-controlling of X6CrNiMoVNb11-2 steel need to be investigated further.

## 4. Conclusions

In this paper, the alloy phase diagram and CCT curve of X6CrNiMoVNb11-2 stainless steel have been calculated, and the phase transformation and microstructure after various heat treatments have been analyzed. In addition, the mechanical properties after the quenching and tempering process are discussed, and the tempering hardness equation is determined. The results can be summarized as follows:According to the thermal expansion curve, the A_c1_, A_c3_, M_s_ and M_f_ points are determined to be A_c1_ = 774.77 °C, A_c3_ = 851.21 °C, M_s_ = 292.71 °C, M_f_ = 133.95 °C, and the tempering hardness equation under the quenching-tempering process is obtained;Secondary hardening occurs at the tempering temperature of 550 °C, due to the formation of M_7_C_3_ in martensite. When the tempering temperature increases up to 650 °C, the M_7_C_3_ carbides become coarser and begin to partially convert into M_23_C_6_ carbides;During the tempering process, the cooling rate has a greater impact on the impact toughness. The faster the cooling rate is, the higher the impact toughness is. With the continuing phase transformation from austenite into martensite that depletes the carbon element, the hardness greatly increases and the elongation slightly decreases;The grain size of martensite is finer and more uniform under the process of the oil quenching and tempering with air cooling, compared with that of the oil quenching process. The optimized process, namely, oil quenching at 1040 °C for 1.0 h and then tempering at 650 °C for 1.5 h with the air cooling, yields the best comprehensive properties;The ideal heat treatment is to quench at 1040 °C with oil cooling and then temper at 650 °C with air cooling, resulting in hardness on the surface of 313 HV, the hardness at the center is 285 HV, the elongation is 17.9%, and the impact toughness is 65.0 J/cm^2^;The cooling medium has an important effect on mechanical properties. With the decrease in the cooling rate, carbides gradually precipitate during quenching, and the subsequent temper embrittlement appears during tempering, resulting in a great decrease in impact toughness and strength. Thus, high impact toughness can be obtained through rapid cooling during the quenching and tempering processes.

In brief, this basic investigation of the phase transformation kinetic equation, microstructure, phase precipitation and mechanical properties after various heat treatments help to reveal the reinforcement mechanisms and performance-controlling scheme of a martensite X6CrNiMoVNb11-2 gas turbine disc. The optimized deformation parameters, such as dynamic recovery, dynamic recrystallization, superplastic deformation, cracks, and local flow stress will be further investigated in the future.

## Figures and Tables

**Figure 1 materials-14-05243-f001:**
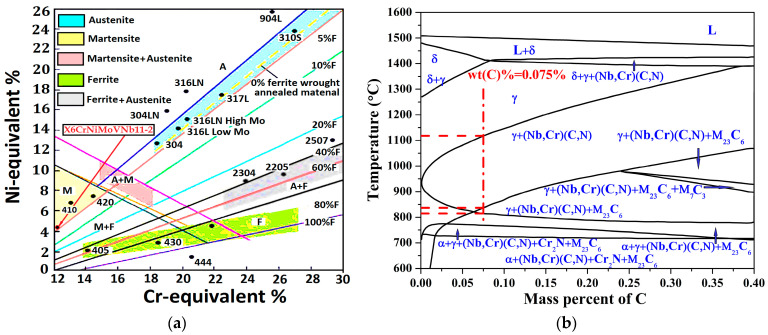
Schaeffler diagram and equilibrium phase diagram: (**a**) Schaeffler diagram; (**b**) equilibrium phase diagram.

**Figure 2 materials-14-05243-f002:**
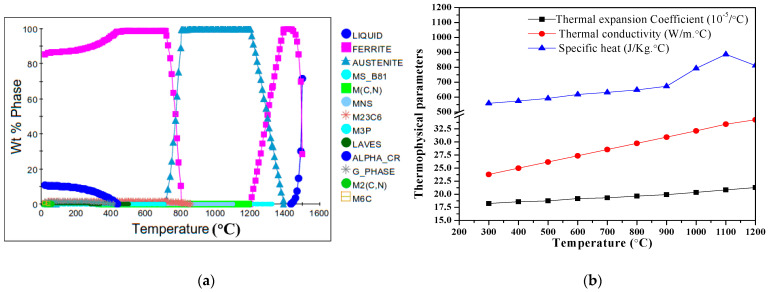
Solidification of X6CrNiMoVNb11-2 steel: (**a**) alloy phase diagram; (**b**) thermophysical parameters.

**Figure 3 materials-14-05243-f003:**
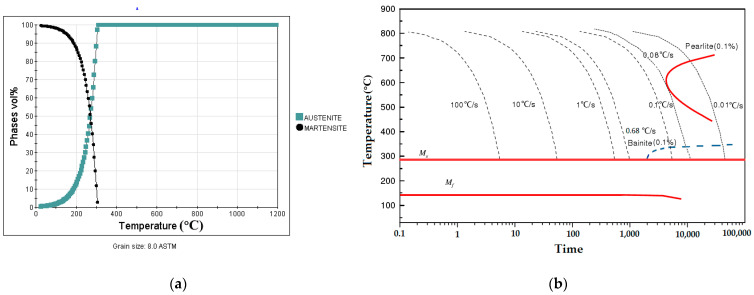
Calculated curves of X6CrNiMoVNb11-2 steel using the JMatPro database: (**a**) relationship between the phase composition and the cooling temperature at the cooling rate of 1 °C/s; (**b**) critical cooling rate curves.

**Figure 4 materials-14-05243-f004:**
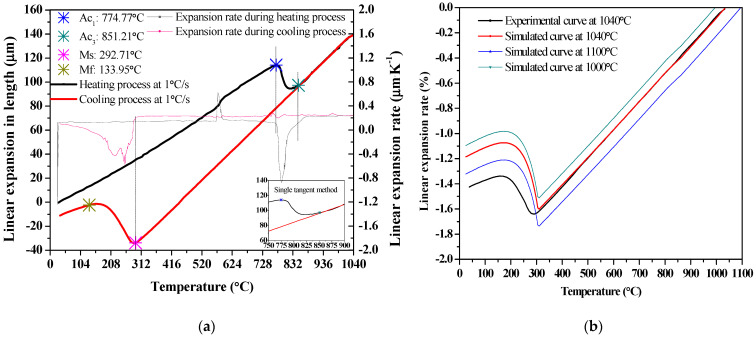
Linear expansion with a cooling rate of 1.0 °C/s: (**a**) experimental curve; (**b**) comparison of simulated and experimental curves.

**Figure 5 materials-14-05243-f005:**
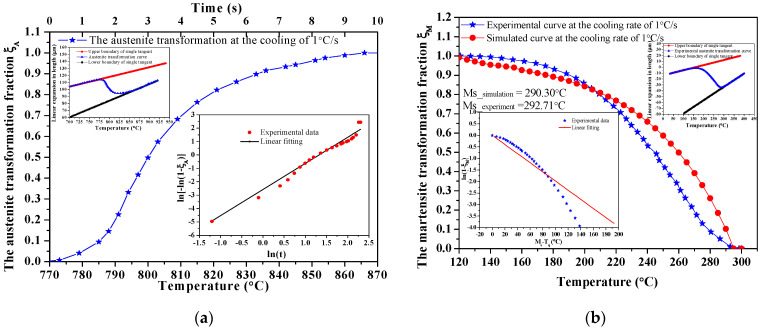
Phase transformation fraction in cooling at 1.0 °C/s: (**a**) austenite transformation; (**b**) martensite transformation.

**Figure 6 materials-14-05243-f006:**
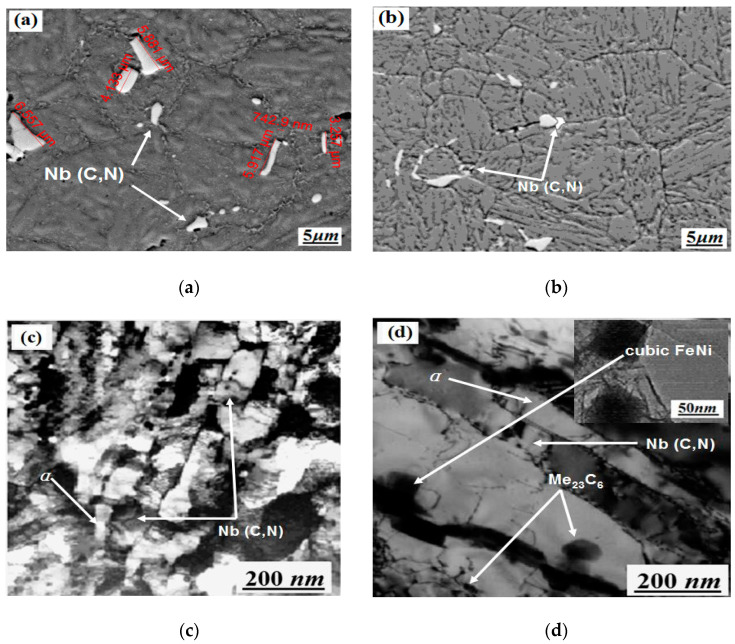
SEM analysis of X6CrNiMoVNb11-2 steel before and after heat treatment: (**a**) SEM of the initial as-rolled microstructure before heat treatment; (**b**) SEM after quenching (at 1040 °C, oil cooling) and tempering (at 650 °C, oil cooling) processes; (**c**) TEM of the initial as-rolled microstructure before heat treatment; (**d**) TEM after quenching (at 1040 °C, oil cooling) and tempering (at 650 °C, oil cooling) processes.

**Figure 7 materials-14-05243-f007:**
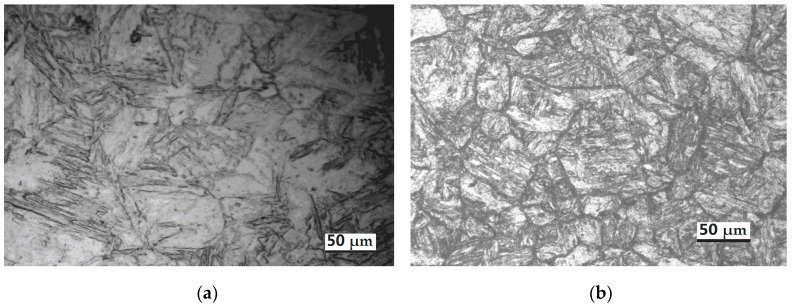
OM and XRD analysis using different quenching cooling methods: (**a**) quenching at 1040 °C with oil cooling; (**b**) quenching at 1040 °C with air cooling and tempering at 650 °C in air cooling; (**c**) quenching at 1040 °C with oil cooling and tempering at 650 °C in air cooling; (**d**) comparison of XRD under various heat treatment processes.

**Figure 8 materials-14-05243-f008:**
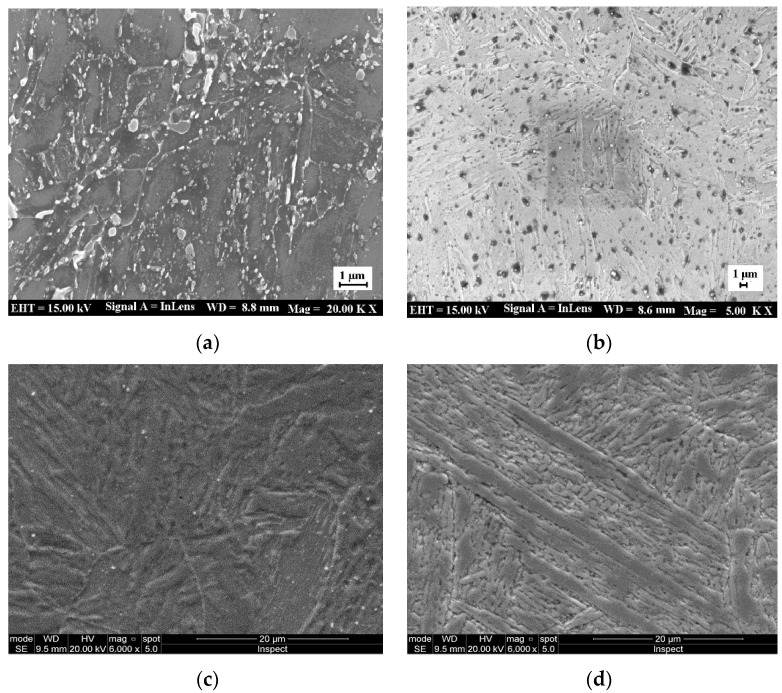
SEM-EDS analysis of X6CrNiMoVNb11-2 steel before and after heat treatment: (**a**) SEM-EDS before heat treatment; (**b**) SEM-EDS after the quenching and tempering processes; (**c**) α + Nb(C, N) by quenching; (**d**) α + Nb(C, N) + M_23_C_6_ by quenching and tempering.

**Figure 9 materials-14-05243-f009:**
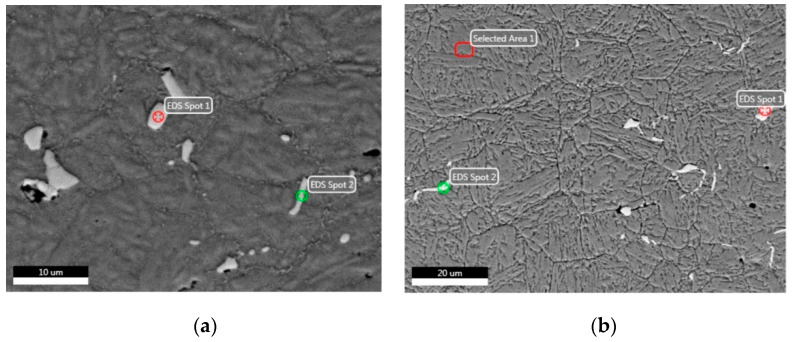
EDS analysis of X6CrNiMoVNb11-2 steel before and after heat treatment: (**a**) EDS of the initial as-rolled X6CrNiMoVNb11-2 steel; (**b**) EDS after quenching (at 1040 °C, oil cooling) and tempering (at 650 °C, oil cooling) processes.

**Figure 10 materials-14-05243-f010:**
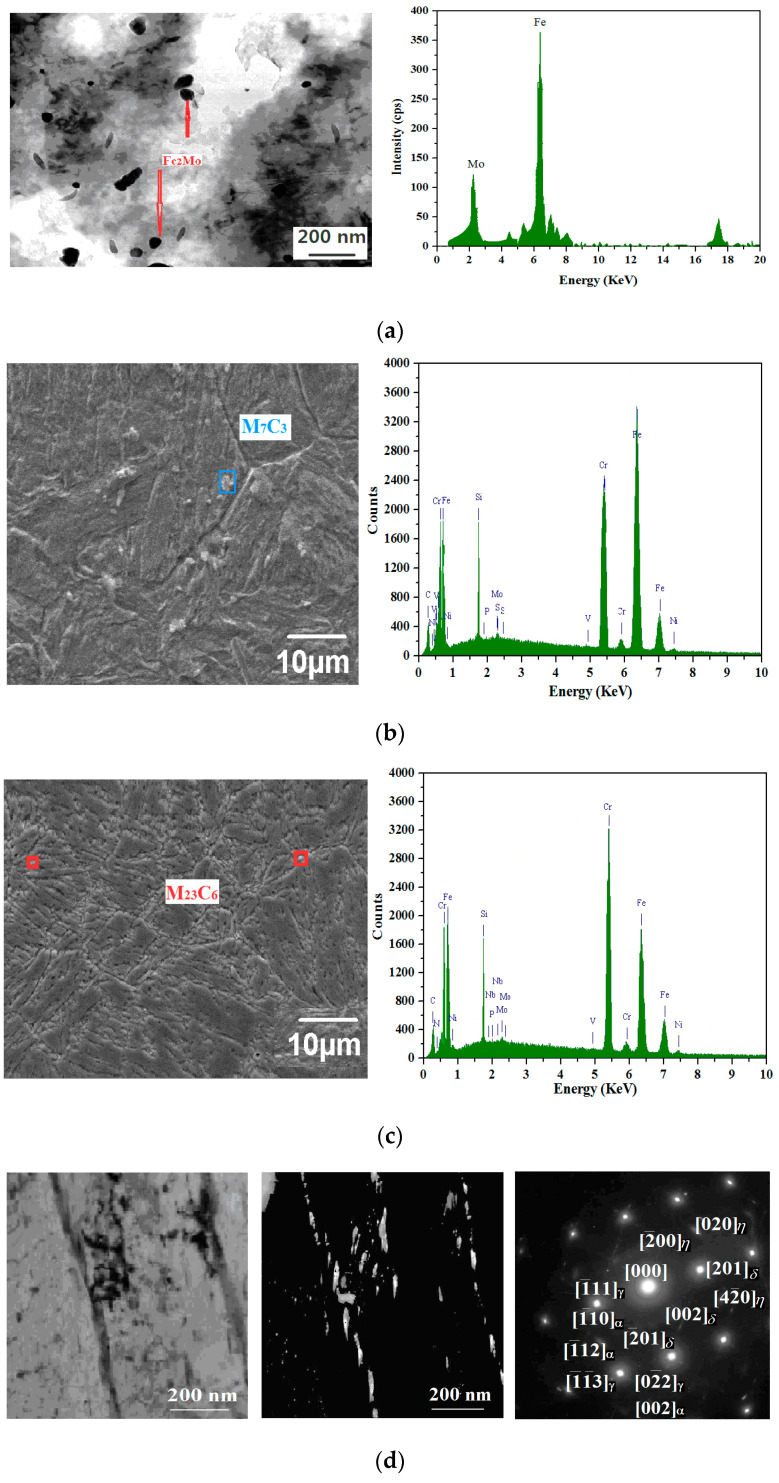
Phase analysis of the microstructure after various heat treatments: (**a**) TEM bright field image of the Laves-Fe_2_Mo phase after the quenching process; (**b**) microstructure consisting of α + Nb(C, N) + M_7_C_3_ by quenching and tempering at 550 °C; (**c**) microstructure of α + Nb(C, N) + M_23_C_6_ by quenching and tempering at 650 °C. (**d**) Bright-field image, dark-field image, and diffraction pattern from quenching and tempering at 650 °C.

**Figure 11 materials-14-05243-f011:**
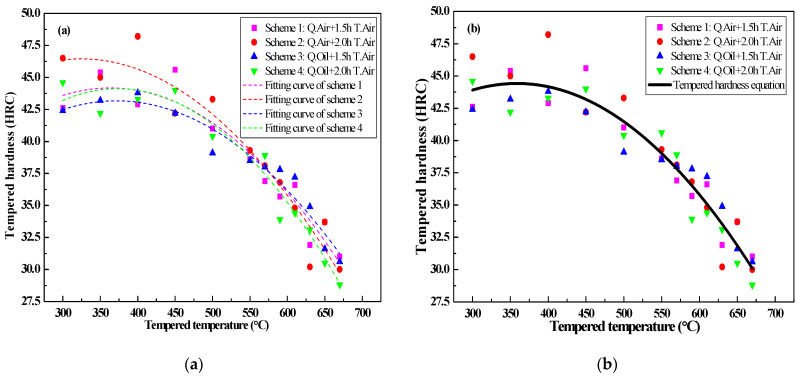
Tempering hardness equation: (**a**) hardness-temperature fitting curves; (**b**) tempered hardness curves.

**Figure 12 materials-14-05243-f012:**
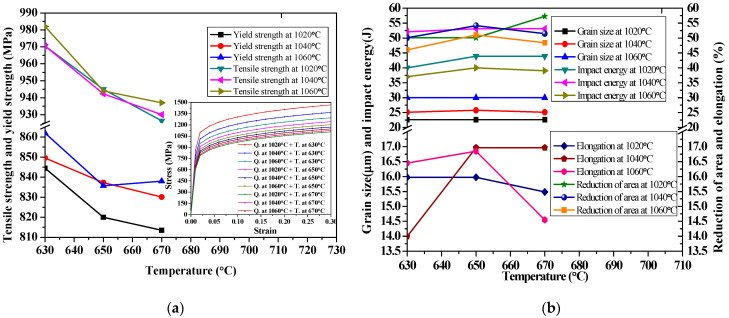
Measured mechanical properties: (**a**) tensile and yield strengths; (**b**) grain size, impact energy, reduction of area and elongation.

**Table 1 materials-14-05243-t001:** Tensile properties and hardness of as-cast and as-rolled X6CrNiMoVNb11-2 steel.

	Tensile Properties					Hardness
	σ_b_ (MPa)	σ_0.2_ (MPa)	δ (%)	Ψ (%)	A_k_ (J/cm^2^)	HV
As-cast	815	695	8.3	45.0	45.0	238
As-rolled (Initial)	925	735	12.1	46.0	55.0	328

**Table 2 materials-14-05243-t002:** Schedule of the heat treatment experiment of X6CrNiMoVNb11-2 stainless steel.

Quenching Process			Tempering Process		
Temperature	Time	Cooling	Temperature	Time	Cooling
		Oil cooling			
1040 °C	1 h	Air cooling	300–670 °C	1.5–2 h	Oil cooling
		Furnace cooling			Air cooling
		PAG medium			

**Table 3 materials-14-05243-t003:** EDS spot analyses of X6CrNiMoVNb11-2 steel before and after heat treatment.

Spot Position	Element	Weight %	Atomic %	Net Int.	Spot Position	Element	Weight %	Atomic %	Net Int.
Spot 1 in Figure 9a	NbL	91.46	86.20	1472.15	Spot 1 in Figure 9b	NbL	93.01	88.56	982.32
	TiK	0.67	1.22	10.64		V K	1.14	1.97	10.38
	CrK	2.01	3.38	26.53		CrK	1.63	2.77	14.16
	FeK	5.87	9.20	62.60		FeK	4.22	6.69	29.70
Spot 2 in Figure 9a	NbL	69.09	57.02	1051.50	Spot 2 in Figure 9b	NbL	86.85	79.54	843.12
	TiK	0.17	0.28	2.93		V K	1.02	1.71	8.87
	CrK	4.98	7.35	69.55		CrK	2.34	3.82	19.21
	FeK	25.75	35.35	273.78		FeK	9.79	14.92	64.28

**Table 4 materials-14-05243-t004:** Phases of X6CrNiMoVNb11-2 martensitic steel after various heat treatments.

Regime of Heat Treatment	Hardness (HV)	Phase Composition
Quenching at 1040 °C by oil	380.0–420.0	α + Nb(C, N)
Quenching at 1040 °C by oil + tempering at 550 °C	375.0–395.0	α + Nb(C, N) + M_7_C_3_ + others [48]
Quenching at 1040 °C by oil + tempering at 650 °C	300.0–330.0	α + Nb(C, N) + M_23_C_6_ + others [48]

**Table 5 materials-14-05243-t005:** Effect of quenching cooling rate on mechanical properties at 1040 °C during quenching and tempering.

	Outer Layer				Center Position			
	Oil. + Air.	Air. + Air.	Fur. + Air.	Con. + Air.	Oil. + Air.	Air. + Air.	Fur. + Air.	Con. + Air.
Tensile strength (MPa)	1045	1060	1050	1020	1052	1060	1050	1020
Yield strength (MPa)	962	948	933	916	975	946	933	910
Impact toughness (J/cm^2^)	65.0	58.8	56.8	55.0	64.0	58.8	57.9	54.5

**Table 6 materials-14-05243-t006:** Effect of tempering cooling rate on mechanical properties at 650 °C during quenching and tempering.

	Outer Layer			Center Position		
	Oil. + Oil.	Oil. + Air.	Oil. + Fur.	Oil. + Oil.	Oil. + Air.	Oil. + Fur.
Tensile strength (MPa)	944	951	941	941	948	939
Yield strength (MPa)	835	853	830	830	837	826
Impact Toughness (J/cm^2^)	95.0	67.0	57.5	92.0	65.0	45.0

**Table 7 materials-14-05243-t007:** Hardness of X6CrNiMoVNb11-2 steel by three tempering conditions at 650 °C after quenching and tempering.

Quenching and Tempering Medium	Surface Hardness	Central Hardness	Difference between Surface and Center
Air	308 HV	279 HV	29 HV
Oil	313 HV	285 HV	27 HV
PAG medium	335 HV	297 HV	38 HV
Oil (quenching) + air (tempering)	340 HV	318 HV	22 HV

**Table 8 materials-14-05243-t008:** Comparison of tensile properties, hardness and the subsequent phases under different heat treatment processes.

Condition	σ_b_ (MPa)	σ_0.2_ (MPa)	δ (%)	Ψ (%)	Ak(J/cm^2^)	Phase	Hardness (HV)
Quenching by oil (Q.Oil)	952.5	853.5	12.0	44.5	65.0	α + Nb(C, N)	418
Quenching by air (Q.Air)	965.0	838.0	11.9	44.5	65.0	α + Nb(C, N)	408
Q.Oil + 550 °C tempering	951.0	884.0	15.5	48.0	70.0	α + Nb(C, N) + M_7_C_3_	392
Q.Air + 550 °C tempering	993.0	850.0	14.2	46.0	64.0	α + Nb(C, N) + M_7_C_3_	382
Q.Oil + 650 °C tempering	910.5	885.5	17.9	55.0	65.0	α + Nb(C, N) + M_23_C_6_	313
Q.Air + 650 °C tempering	925.0	820.0	16.8	53.0	60.5	α + Nb(C, N) + M_23_C_6_	308
Q.Oil + 670 °C tempering	910.0	810.5	16.5	52.0	62.0	α + Nb(C, N) + M_23_C_6_ (coarser)	315
Q.Air + 670 °C tempering	920.0	805.5	15.5	49.0	60.0	α + Nb(C, N) + M_23_C_6_ (coarser)	298

## Data Availability

Data available in a publicly accessible repository that does not issue DOIs. Publicly available datasets were analyzed in this study.

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
