# Peer review of "Microstructure Evolution and Mechanical Properties of X6CrNiMoVNb11-2 Stainless Steel after Heat Treatment"

_materials, 2021, doi:10.3390/ma14185243_

Round 1
Reviewer 1 Report
My recommendation is minor revision taking into account the following ;
1 introducing a detailed discussion of XRD spectra
2 correct errors such as absence of the letter n in the word “quenching “( see line 29,342,362,486 )
Author Response
1 introducing a detailed discussion of XRD spectra
Thank you for your objective suggestion. We have introduced a comparison of XRD spectra under various heat treatments processes in section 3.3.1. Which is as follows:
3.3.1 SEM and XRD analysis before and after heat treatment
Taking the quenching at 1040 °C for 1.0 h + tempering for 1.5 h at 650 °C air cooling as an example, microstructures under various quenching methods are in Figure 7.
|
(a) |
(b) |
|
(c) |
(d) |
Figure 7. OM and XRD analysis using different quenching cooling methods: (a) Quenching at 1040 °C with oil cooling; (b) Quenching at 1040 °C with air cooling and tempering at 650°C in air cooling; (c) Quenching at 1040 °C with oil cooling and tempering at 650°C in air cooling; (d) Comparison of XRD under various heat treatments processes.
It can be seen from Figure 7a that the quenching microstructure with oil cooling and quenching is mainly lath martensite and tempering processes in Figure 7b and Figure 7c are mainly tempered martensite + carbide. The microstructure under process in Figure 7c (quenching oil cooled at 1040 °C and tempered at 650 °C for 1.5 h) is more uniform. Compared with the quenching air-cooled structure in Figure 6b, the quenching oil-cooled structure in Figure 6b is smaller and more uniform due to the different cooling rates. The XRD curves under various processes shows that the martensite is mainly distributede and the content of retained austenite is almost zero. After heat treatment, the average austenite diameter is 25 µm, corresponding to ASTM-8 grain size.
2 correct errors such as absence of the letter n in the word “quenching “( see line 29,342,362,486 )
We noticed it and made the change as suggested.

Reviewer 2 Report
Thank you for submitting this work to "Materials" - I think this is an excellent fit.
Here some more suggestions that can further improve the quality of your work:
-Abstract: good - check sentence structure line 20-24
- I guess this is a research article - please mark it as such - top of 1st page
- your work is not only important for tempering or heat treatment but also diffusion bonding (https://doi.org/10.3390/met10050613) that essentially includes a heat treatment
I think this should be mentioned since the applications (aerospace, other corrosive environments etc) of 06Cr11Ni2MoVNb seem to largely overlap with areas where diffusion bonding is used
Explain line 63-65 better - this is a big statement that needs more details - elaborate a bit what they did in Ref. 15
I think you can quickly provide a paragraph about 06Cr11Ni2MoVNb stainless steel here so that readers know the exact composition of the material
Maybe there are also EU and American names for it? I think this would be helpful for readers outside China and probably also within China since it seems to be a rather exotic material
Is this an of the shelf material or did you produce it yourself in the lab? I think this should be mentioned
How do you know the initial properties in Table 1? Did you measure that? Since you seemed to have produced the material yourself these cannot be values from a vendor - please elaborate a bit
Fig 1 - highlight better where your material is in here
Fig 1-2, and others try to use the same style of your figures - fonts, etc.
Fig. 11 very good - this is really helpful
The results and discussion session and also the conclusions are well done.
Really try to explain better if this is your material or you try to reproduce something that can be bought in China.
This is not clear to me yet.
Author Response
Thank you for submitting this work to "Materials" - I think this is an excellent fit.
Here some more suggestions that can further improve the quality of your work:
-Abstract: good - check sentence structure line 20-24
Thank your for your kind suggestion. We noticed it and made the change as foolows:
A certain amount of the high-density nanophase precipitation are found by the martensite phase transformation and heat treatmen, where M23C6 carbides are dispersed in lamellar martensite, with the close-packed Ni3Mo and Ni3Nb phases of high-density colattice nanocrystalline precipitation during the tempering process at 650 °C. The ideal process parameters is to quench at 1040 °C by oil cooling medium and to temper at 650 °C by air cooling, with final hardness averaged about 313 HV, the elongation of 17.9%, reduction ratio in area 52% and impact toughness about 65 J, respectively.
- I guess this is a research article - please mark it as such - top of 1st page
Thank you very much. We noticed it and made the change as suggested.
- your work is not only important for tempering or heat treatment but also diffusion bonding (https://doi.org/10.3390/met10050613) that essentially includes a heat treatment. I think this should be mentioned since the applications (aerospace, other corrosive environments etc) of 06Cr11Ni2MoVNb seem to largely overlap with areas where diffusion bonding is used
- AlHazaa, N. Haneklaus, Diffusion Bonding and Transient Liquid Phase (TLP) Bonding of Type 304 and 316 Austenitic Stainless Steel—A Review of Similar and Dissimilar Material Joints, Metals2020,10(5), 613.
Thank you for the reminder, and I quote the reference as follows:
For the martensite stainless steel, tempering solid solution treatment process on the has great influence on microstructure and mechanical properties of high strength steel and the grain after two-time tempering process is finer [12]. The increase of the molybdenum content can significantly reduce the corrosion reaction rate [13]. The mechanical toughness after the solid solution at 1040 °C for 1 h and two-time tempering at 600 °C for 3 hours meets the requirement of 00Cr13Ni4Mo steel in application[14]. In fact, the applications of the aerospace and other corrosive environments of X6CrNiMoVNb11-2 seem to largely overlap with areas where diffusion bonding is used [15]. The lath martensite and a small amount of retained austenite after quenching at 800 - 1100°C for 0.5h for 13Cr steel [16]. Besides, with the increase of austenitizing temperature, the martensite lath and retained austenite grains might be coarsened [17-18]. Therefore, the proper quenching and tempering process of low carbon martensitic stainless steel after the austenitic stabilization treatment becomes meaningful in practical applications [6-8], especially for the X6CrNiMoVNb11-2 steel with excellent hardenability.
References
- Bilmes, P. D., Solari, M., Llorente, C. L. Characteristics and effects of austenite resulting from tempering of 13CrNiMo martensitic steel weld metals. Charact. 2011, 46, 285-296.
- Kondo, K., Ogawa, K., Amaya, H., Ueda, M., Ohtani, H. Development of weldable super 13Cr martensitic stainless steel for flowline. Conf. the Twelfth Int. Offshore and Polar Engineering (Kitakyushu, Japan) 2002, 26–31.
- Zou, D. , Han, Y., Zhang, W., Fang, X. D. Influence of tempering process on mechanical properties of 00Cr13Ni4Mo supermartensitic stainless steel. J. Iron. Steel Res. Int. 2010, 17, 50-54.
- AlHazaa, N. Haneklaus, Diffusion Bonding and Transient Liquid Phase (TLP) Bonding of Type 304 and 316 Austenitic Stainless Steel—A Review of Similar and Dissimilar Material Joints, Metals2020, 10 613.
Explain line 63-65 better - this is a big statement that needs more details - elaborate a bit what they did in Ref. 15
In fact, the conventional quenching methods are not entirely suitable for steels with high chromium content to improve the microstructure and stress corrosion resistance[15].
- Kushida, T., Kudo, T. Effect of Chromium, Molybdenum and nickel on hydrogen embrittlement on martensitic steels. Zairyo-to-Kankyo (Corrosion Engineering) 1992, 41 677-683.
Thanks for your suggestion. Since there is little correlation between heat treatment performance and corrosion, I have omitted the description in this section for the sake of avoiding ambiguity.
I think you can quickly provide a paragraph about 06Cr11Ni2MoVNb stainless steel here so that readers know the exact composition of the material
Thank you. I've reorganized the sentences about the usage of this steel and the research background in the introduction part.
X6CrNiMoVNb11-2 steel (06Cr11Ni2MoVNb in the CN system) as its high-strength, good hardenability, creep resistance and corrosion-resistance martensite stainless steel used in aero engine turbine discs, large gas turbines, turbine blades, nuclear power equipment, corrosion of oil pipelines and others……
In previous work, the precise constitutive model under consideration of strain compensation and the hot processing map of X6CrNiMoVNb11-2 steel (06Cr11Ni2MoVNb in the CN system) has been established to distinguish the "safety zone" from the "unstable zone" [19-20]. The hot deformation parameters of ring rolling processing are optimized to investigate the deformation rules of disk piece [21]. However, for this kind of Fe-Cr-Ni-Mo martensitic stainless steel, the main heat treatment process mentioned is only focus on the conventional annealing, quenching and tempering methods, nitriding technology and cooling rate on impact toughness [21,21-26], the influence of quenching and tempering cooling medium on the yield strength, tensile strength, elongation and impact toughness has not revealed yet……
3.1 Analysis of the alloy phase diagram and TTT/CCT curve
Chemical composition was: C: 0.075%, Cr: 11.390 %, Ni: 1.5700 %, Mo: 0.390 %, Nb: 0.080 %, V: 0.150 %, Mn: 0.380 %, Si: 0.180 %, N: 0.02 %, P: 0.015 %, S: 0.004 %, other: Fe.
Maybe there are also EU and American names for it? I think this would be helpful for readers outside China and probably also within China since it seems to be a rather exotic material
Is this an of the shelf material or did you produce it yourself in the lab? I think this should be mentioned
Thank you very much. In fact, this material of 06Cr11Ni2MoVNb in the CN system is equal to X6CrNiMoVNb11-2 in the EN system. It can be obtained by either the as- cast product from the laboratory or the ordered product from Northeast Special Steel Co., Ltd. I have indicated in the revised version to mark that it is produced by this company.
How do you know the initial properties in Table 1? Did you measure that? Since you seemed to have produced the material yourself these cannot be values from a vendor - please elaborate a bit
Thank you for your suggestion, the initial performance is provided by the supplier, and I have noted in the revised version as follows:
2.1.2 Hot rolling process and the initial as-rolled mechanical properties
The pre-heat treatment before rolling consisted of steps as: 1) homogenizing the steel: heating to 700 °C and holding it for 5 - 6 h in the box resistance furnace, then cooling it at room temperature; 2) normalizing the steel: heating it to 1040 °C and holding it for 1 - 2 h, then cooling it at room temperature. The scaling test, the ring of cast steel ingot (50 mm inner diameter, 280 mm outer diameter and 32 mm in height) was heated to 1250 °C and homogenized for 4 h, then cooled to 1200 °C for 1 h to approach the initial state of hot rolling after casting, a special cone roll die was loaded on the ring rolling tester to carry out isothermal hot rolling. The size of the rolled plate is 135 mm inner diameter, 386 mm outer diameter and 28 mm in height. After pre-heat treatment and rolling, the initial properties of as-rolled disk measured from section 2.2.2, were listed in Table 1.
Table 1. Tensile properties and hardness of as-cast and as-rolled X6CrNiMoVNb11-2 steel.
|
|
Tensile properties |
Hardness |
||||
|
|
σb(MPa) |
σ0.2 (MPa) |
δ (%) |
ψ(%) |
Ak(J/cm2) |
HV |
|
As-cast |
815 |
695 |
8.3 |
45.0 |
45.0 |
238 |
|
As-rolled (Initial) |
925 |
735 |
12.1 |
46.0 |
55.0 |
328 |
Table 1 shows the tensile strength, yield strength, elongation and impact toughness of as-rolled specimens are increased by 110MPa, 40MPa, 3.8% and 10J/cm2, compared with as-cast properties provided by the vendor of Northeast Special Steel Co. Ltd. The initial as-rolled specimens were used for heat treatment, detailed in section 2.2.
Fig 1 - highlight better where your material is in here
Thank you very much for your suggestion, the chemical composition is highlighted here in Fig.1 to obtained the chromium and nickel equivalent, shown in section 3.1.
3.1 Analysis of the alloy phase diagram and TTT/CCT curve
Chemical composition was: C: 0.075%, Cr: 11.390 %, Ni: 1.5700 %, Mo: 0.390 %, Nb: 0.080 %, V: 0.150 %, Mn: 0.380 %, Si: 0.180 %, N: 0.02 %, P: 0.015 %, S: 0.004 %, other: Fe. Thus, the chromium and nickel equivalent by Schaeffler diagram are calculated as [11]:
|
CrEqu=%Cr+%Mo+1.5×%Si+0.5×%Nb=11.39%+0.39%+1.5×0.18%+0.5×0.08%=12.09% |
(1) |
|
NiEqu=%Ni+30×%C+0.5×%Mn=1.57%+30×0.075%+0.5×0.38%=4.01% |
(2) |
Where, CrEqu and NiEqu are chromium and nickel equivalent, respectively.
Fig 1-2, and others try to use the same style of your figures - fonts, etc.
Thank you very much for your notice. We authors check the whole text to make sure the standard writing and the unit’s spacing in each section.
Fig. 11 very good - this is really helpful
Thanks again.
The results and discussion session and also the conclusions are well done.
Really try to explain better if this is your material or you try to reproduce something that can be bought in China.
This is not clear to me yet.
Thank you for your question. This material of 06Cr11Ni2MoVNb in the CN system is equal to X6CrNiMoVNb11-2 in the EN system. This steel is produced by the Northeast Special Steel Co., Ltd. It can be bought in China. I have indicated in the revised version to mark that it is produced by this company.
Besides, the introduction, the research design and the methods have been improved, with the red marked part in the revised manuscript this time.
Thanks again.

Reviewer 3 Report
The present manuscript deals with microstructure evolution and mechanical properties of heat-treated 06Cr11Ni2MoVNb stainless steel. This manuscript includes engineering and scientifically interesting results. However, major revision is needed to be accepted as an article in Metals.
Reviewer’s comments are as follows,
- Structure of this paper
- This paper investigates the microstructure evolution and mechanical properties of 06Cr11Ni2MoVNb stainless steel. However, the authors do not explain important results of mechanical properties of the steel (Tables 5 to 8 and Figure 11a) in Result. Please move the results (Tables 5 to 7 and Figure 11a) in the Discussion.
- The results are included in the Discussion. Please divide the results and discussion.
- significant figures
Please round to three or four significant figures.
- Dimensions of tensile specimen
Please show the thickness of the tensile specimen.
- The cooling rate in the quenching process
Table 2: What quenching media is a cooling rate of 1.0 °C/mm?
Table 7: What are “PAG types” and “Water-air”. Please explain them in detail.
- Figure 6
Please add the kind of carbide and heat treatment condition in this caption.
- L295-L296
We cannot find Figures 8c and 8d.
- L363 and L364
- Please explain about “Con”.
- We can see a cooling rate of 1.0 C/min in Table 2. However, the rate is 10 C/min in L364. Why?
That’s all
Author Response
Reviewer’s comments are as follows,
- Structure of this paper
- This paper investigates the microstructure evolution and mechanical properties of 06Cr11Ni2MoVNb stainless steel. However, the authors do not explain important results of mechanical properties of the steel (Tables 5 to 8 and Figure 11a) in Result. Please move the results (Tables 5 to 7 and Figure 11a) in the Discussion.
Thank you for your suggestions. The mechanical properties of the steel (Tables 5 to 8 and Figure 11a) has been move from the results (Tables 5 to 7 and Figure 11a) into the Discussion section, seen in the revised version.
- The results are included in the Discussion. Please divide the results and discussion.
Thank your for your effective proposal. The structure of the paper is reorganized according to your suggestions, which is as follows:
- Results
3.1 Analysis of the alloy phase diagram and TTT/CCT curve
3.1.1 Analysis of the alloy phase diagram
3.1.2 Analysis of the phase composition and CCT curves
3.2 Phase transformation temp eratures and martensitic kinetic equation
3.2.1 Determination of martensitic phase transition point
3.2.2 Calculation of phase transformation kinetic equation
3.3 Analysis of SEM, XRD, EDS and phase before and after heat treatment
3.3.1 SEM and XRD analysis before and after heat treatment
3.3.2 SEM-EDS analysis before and after heat treatment
3.4 Phase analysis before and after heat treatment
3.4.1 EDS analysis before and after heat treatment
3.4.2 Phase analysis before and after heat treatment
3.4.3 Hardness analysis after various heat treatment processes
- Discussion
4.1 Experimental mechanical properties under various heat treatments
4.1.1 Effect of quenching cooling speed on properties
4.1.2 Effect of tempering cooling speed on properties
4.1.3 Comparison of hardness under various processes
4.2 Comparison of mechanical properties under various processes
4.2.1 Determination of the tempering hardness equation
4.2.2 Comparison of tensile properties under various processes
- significant figures
Please round to three or four significant figures.
Thank your for your effective proposal. Figure 7, Figure 8 and Figure 10 are rounded to four significant figures in the revised version.
- Dimensions of tensile specimen
Please show the thickness of the tensile specimen.
Thanks. The thickness of the tensile specimen is added.
2.2.2 Experimental measurement of mechanical properties
The DIL805A/D differential dilatometer is used to measure the linear expansion curve of sample (Φ4 mm × 10 mm) and record the relative displacement to determine the temperature at the characteristic points of phase transition, such as Ms, Mf, Ac1 and Ac3. Besides, the Charpy notched specimen for toughness measurement was tested by the V-notched with the size of 5 mm × 10 mm × 55 mm (GB/T229-1994). Specimen tensile strength was tested with the flat section size seen in ASTM-E8M (the specifications are as: gauge length is 50.0 ± 0.1 mm, contraction width is 12.5 ± 0.2 mm, thickness is 4.0 mm, bending radius ≥12.5 mm, contraction length ≥60 mm, clamping length ≥75 mm, clamping width 20 mm). Microhardness was measured by vickers hardness tester under the load of 200 gf, dwell time about 20 s, and spacing between indentations about 0.1 mm. Furthermore, each experiment was carried out three times and the experimental data was averaged. Specimens were polished and etched with the etchant (4ml HF+ 4ml HNO3 + 92ml distilled water) about 5 minutes for each specimem to reveal the microstructures. The phase, microstructures were observed and analyzed by means of XRD-6000 X-ray diffractometer, VEX-600E optical microscope(OM), JSM-7100F scanning electron microscope (SEM) and energy dispersive spectroscopy (EDS). The hardness, elongation and impact toughness were carried out on the FM-700 FM-ARS microhardness tester, PLT-10 slow strain rate tensile tester and MTS universal tester, separately.
- The cooling rate in the quenching process
Table 2: What quenching media is a cooling rate of 1.0 °C/mm?
Table 7: What are “PAG types” and “Water-air”. Please explain them in detail.
Thanks for your suggestion.
(1) In Table 2, the 1.0 °C/s is correct in Table2, instead of the previous wrong expression of “1.0 °C/mm”.
(2) In Table 7, “PAG types” and “Water-air” is described as “PAG medium (about 1°C /s)” and “Water (quenching) + air (tempering)” in the revised version, and the Table 7 is updated as follows:
Table 7. Hardness of X6CrNiMoVNb11-2 steel by three tempering conditions at 650 °C after quenching and tempering.
|
Quenching and tempering medium
|
Surface hardness HV |
Central hardness HV |
Difference between surface and center HV |
|
Air |
308 |
279 |
29 |
|
Oil |
313 |
285 |
27 |
|
PAG medium (about 1°C /s) |
335 |
297 |
38 |
|
Water (quenching) + air (tempering) |
340 |
318 |
22 |
- Figure 6
Please add the kind of carbide and heat treatment condition in this caption.
The carbide and heat treatment condition in Fig.6 are added in the revised version.
Figure 6. SEM analysis of X6CrNiMoVNb11-2 steel before and after heat treatment: (a) Initial as-rolled microstructure before heat treatment; (b) SEM after quenching (at 1040°C, oil cooling) and tempering (at 650°C, oil cooling) processes.
- L295-L296
We cannot find Figures 8c and 8d.
The Figures 8c and 8d are added this time, in the revised version. Thanks again.
Figure 8. SEM-EDS analysis of X6CrNiMoVNb11-2 steel before and after heat treatment: (a) SEM-EDS before heat treatment; (b) SEM-EDS after quenching and tempering processes; (c) α + Nb(C, N) by queching; (d) α + Nb(C, N) + M23C6 by queching and tempering.
- L363 and L364
- Please explain about “Con”.
- We can see a cooling rate of 1.0 C/min in Table 2. However, the rate is 10 C/min in L364. Why?
Thank you for your questions.
- “Con.” stands for the constant-speed cooling at 1.0°C/min.
2.There's a punctuation missing. It's been corrected in the revised revision.
Besides, the research design, the methods and the results have been improved, with the red marked part in the revised manuscript this time.
Thanks again.

Round 2
Reviewer 2 Report
Good work - all issues have been thoroughly addressed. This work can now be published.
Author Response
Thank you. We authors check the whole text, English language and style to verify and to make sure the standard writing and the unit’s spacing in each section in the revised version.
Reviewer 3 Report
Reviewer’s comments are as follows,
(1) Structure of this paper
In “4. Discussion”, the results of the mechanical properties are mainly shown with some discussion. Therefore, the present “4. Discussion” is corresponding to “Results and Discussion”. Moreover, “4. Discussion” does not include the discussion on microstructure at all. Therefore, I recommend that “3. Results and 4. Discussion” are changed into “3. Results and Discussion”. For example,
- Results and Discussion
- Discussion --> delete
4.1 --> 3.5
4.2 --> 3.6
- Conclusions à 4. Conclusions
(2) Table 3
Please explain the abbreviation “PAG”.
(3) P. 4, L2
Dimensions of tensile specimen
Please show the thickness of tensile specimen.
(4) Figure 6a
Please show the kind of carbides.
That’s all
Author Response
(1) Structure of this paper
In “4. Discussion”, the results of the mechanical properties are mainly shown with some discussion. Therefore, the present “4. Discussion” is corresponding to “Results and Discussion”. Moreover, “4. Discussion” does not include the discussion on microstructure at all. Therefore, I recommend that “3. Results and 4. Discussion” are changed into “3. Results and Discussion”. For example,
- Results and Discussion
- Discussion --> delete
4.1 --> 3.5
4.2 --> 3.6
- Conclusions à 4. Conclusions
Thank you. We have made the change as suggested in the revised version.
(2) Table 3
Please explain the abbreviation “PAG”.
Thank you very much. The abbreviation “PAG” is the short of the polyalkylene glycol (PAG) and we have added in the content in the revised version.
(3) P. 4, L2
Dimensions of tensile specimen
Please show the thickness of tensile specimen.
Thanks again. The thickness of tensile specimen is 4mm, and we have marked in the content.
(4) Figure 6a
Please show the kind of carbides.
Many thanks. We have made this change in Fig.6(a).